# Validity of Clinical Symptoms Score to Discriminate Patients with COVID-19 from Common Cold Out-Patients in General Practitioner Clinics in Japan

**DOI:** 10.3390/jcm10040854

**Published:** 2021-02-19

**Authors:** Shiro Sonoda, Jin Kuramochi, Yusuke Matsuyama, Yasunari Miyazaki, Takeo Fujiwara

**Affiliations:** 1Kuramochi Clinic Interpark, Utsunomiya, Tochigi 321-0114, Japan; kuramochi59naika@gmail.com; 2Department of Global Health Promotion, Tokyo Medical and Dental University, Tokyo 113-8519, Japan; matsuyama-thk@umin.org (Y.M.); fujiwara.hlth@tmd.ac.jp (T.F.); 3Department of Respiratory Medicine, Tokyo Medical and Dental University, Tokyo 113-8519, Japan; miyazaki.pilm@tmd.ac.jp

**Keywords:** severe acute respiratory syndrome coronavirus-2, out-patient, non-COVID-19, clinical symptom

## Abstract

Objective: Coronavirus disease 2019 (COVID-19) has spread worldwide, including Japan. However, little is known about the clinical symptoms which discriminate between COVID-19 and non-COVID-19 among outpatients in general practitioner clinics, which is important for efficient case detection. The aim of this study was to investigate the clinical symptoms to discriminate between COVID-19 and non-COVID-19 cases among outpatients in general practitioner clinics during the second wave of the COVID-19 pandemic in Japan in August 2020. Methods: The records of 360 patients who visited a clinic with suspicion of infectious disease and underwent COVID-19 PCR test between 1 and 14 August 2020 were used. The patients filled out a questionnaire on possible clinical symptoms and transmission routes. Multivariate logistic regression was used to investigate the association between clinical symptoms and COVID-19 status. Results: COVID-19-positive patients were 17 (4.7%). Multiple logistic regression analyses showed that anosmia (odds ratio (OR), 25.94 95% confidence interval (CI), 7.15–94.14; *p* < 0.001), headache (OR, 3.31 95% confidence interval (CI), 0.98–11.20; *p* = 0.054), sputum production (OR, 3.32 CI, 1.01–10.90; *p* = 0.048) and history of visiting an izakaya or bar (OR, 4.23 CI, 0.99–18.03; *p* = 0.051) were marginally significantly associated withbeing COVID-19 positive. This model showed moderate predictive power (area under receiver operating characteristic curve = 0.870 CI, 0.761 to 0.971). Conclusions: We found that anosmia, headache, sputum production, history of visiting an izakaya or bar were associated with COVID-19, which can be used to detect patients with COVID-19 in out-patient clinics in Japan. The findings of this study need to be verified in other clinics and hospitals in Japan and other countries with universal healthcare coverages.

## 1. Introduction

Since December 2019, the severe acute respiratory syndrome coronavirus-2 (SARS-CoV-2), a novel coronavirus which emerged in Wuhan, China, has spread worldwide leading the World Health Organization to declare a pandemic. In Japan, coronavirus disease 2019 (COVID-19) was designated as an infectious disease on 28 January 2020. As Japan’s statutory health insurance system provides universal coverage, it is usual for general practitioners to conduct an initial assessment of patients with symptoms of common cold to identify those who are suspected of having the virus.

Many studies have reported clinical symptoms common in COVID-19 [1]. For example, SARS-CoV-2 infection mainly presents flu-like symptoms such as fever, cough and fatigue [2], and COVID-19 presents anosmia and dysgeusia [3,4,5]. Although the factors that contribute to the severity of COVID-19 are gradually being clarified [6], the factors which discriminate between COVID-19 and non-COVID-19 among outpatients with symptoms of common cold are not clear. Notably, such a discrimination is difficult, because COVID-19 is suggested to have high transmissibility not only after but also before symptom onset [7], and most patients with COVID-19 are classified as mildly symptomatic or asymptomatic [8]. For example, although anosmia is considered to be somewhat specific to COVID-19, it is often present after other viral infections [9]. A previous study reported that anosmia alone cannot discriminate between COVID-19 and non-COVID-19 in general practice, but a combination of symptoms and places visited may be able to do so [10].

Notably, simultaneously differentiating patients with COVID-19 from other patients and improving the diagnostic ability of COVID-19 by first impression is important not only for patients, but also for the safety of medical staff itself and contribution to rapid diagnosis and treatment.

In this study, to detect COVID-19 more easily, we aimed to investigate the clinical characteristics which discriminate COVID-19 by comparing between patients with and without COVID-19 among outpatients showing common cold symptoms who visited a general practitioner clinic.

## 2. Method

### 2.1. Study Design and Setting

This study was performed at the Kuramochi Clinic Interpark, in Utsunomiya City, Tochigi, Japan. Since its opening in October 2015, the clinic has been operating as a special unit for common cold symptoms (SUC) where patients with suspected viral and bacterial infection are seen in isolation rooms. As the other clinics and hospitals in the vicinity do not see patients with fever to avoid nosocomial infection from COVID-19, the SUC examined most of the patients with common cold symptoms.

The medical records of patients who visited the SUC between 1 and 14 August 2020, during the second wave of the COVID-19 pandemic in Japan [11], were collected. The data included demographic information, medical history, exposure to patients with COVID-19, symptoms, signs, number of people around the patient in the past week and the places visited in the past two weeks which were possible transmission routes (interviewed and examined by seven medical doctors). The date of disease onset was defined as the day when any symptom was noted. A confirmed case of COVID-19 was defined as a positive result from real-time RT-PCR assay of nasal swab or saliva specimens. The real time RT-PCR tests were performed at Kotobiken Medical Laboratories (Tokyo, Japan) or BML Inc. (Tokyo, Japan). This study followed the Strengthening the Reporting of Observational Studies in Epidemiology (STROBE) reporting guidelines (Appendix A) and was approved by the Ethics Committee at Tokyo Medical and Dental University.

### 2.2. Possible Factors to Discriminate COVID-19

Demographic data variables were age and sex. Symptom variables were highest body temperature, headache, sore throat, dysgeusia, anosmia, nasal discharge, cough, sputum production, nausea/vomiting, diarrhea, stomachache, fatigue, shortness of breath, joint pain, myalgia and lack of appetite. Questions on the places visited were “in the last two weeks, did you go to the following places: host club or hostess club, nightclub, pachinko parlor (an amusement arcade with hybrid slot and pinball machines where patrons typically spend extended period of time), karaoke (a closed space with users making a loud sound with their voice), izakaya (casual dining and drinking establishment) or bar, amusement arcade, movie theater, theater, live music club, amusement park, hospital, dental clinic, beauty salon, elderly day care center, sports gym, sports club, church, welfare facility, airplane, train, taxi, any other place that people gathered or anywhere outside of the prefecture?” Only age and body temperature were filled out, and other questions were answered by checking yes or no.

### 2.3. Statistical Analysis

First, to check the cut-off of body temperature, continuous and categorized (cut-off: 37.0, 37.5, 38.0, 38.5, 39.0 degrees Celsius) body temperatures were compared between COVID-19 and patients without COVID-19. Then, to investigate the association between demographics, clinical symptoms and transmission routes of COVID-19, multiple logistic regression analysis was conducted. First, simple logistic regression was performed to see the crude association with COVID-19. Further, significant variables in the crude models were used in the multiple logistic regression model, and variables were reduced by backward stepwise regression to maximize area under the receiver–operator characteristic curve (AUC). To develop the score, risk factors that showed a significant association with COVID-19 positive in the multiple logistic regression analysis were selected. To create a formula that can predict COVID-19 score using the selected risk factors, odds ratios (OR) from the multiple logistic regression analysis were used for weighting the risk factors. The weighting system based on a previous study [12] was carried out as follows: the score was tripled when ORs ranged from 2.50 to 3.49, quadrupled when ORs ranged from 3.50 to 4.49 and the score was 26 times when ORs ranged 25.50 to 26.49. However, the cut-off for the ORs may be flexible to increase the AUC. Data were analyzed using STATA version 15.0 (StataCorp. 2017. Stata Statistical Software: Release 15. College Station, TX, USA: StataCorp LLC).

## 3. Results

### 3.1. Characteristics of the Study

Patients aged 20 to 59 accounted for 72.5% (261) of the sample, and there were 17 patients with COVID-19 (4.7%) whose clinical spectrum was 14 moderate, 2 moderate and 1 severe illness based on Coronavirus Disease 2019 (COVID-19) Treatment Guidelines by National Institutes of Health. Overall, the mean (standard deviation (SD)) age of patients in this study was 41.1 (18.8) years; 184 patients (51.1%) were men. The mean (SD) of period from symptom onset to RT-PCR test was 3.4 (6.1) days. The mean (SD) of the number of housemates was 2.4 (2.1), of the number of work colleagues in same place: 15.5 (37.7), of the total number of people met and those met without mask in the past week: 20.9 (29.7) and 2.5 (6.4), respectively. The most common symptoms were fever (283 [78.6%]), fatigue (173 [48.1%]) and headache (152 [42.2%]), and the most visited places were hospital (84 [23.3%]) and outside of their own prefecture, i.e., out of Tochigi prefecture (67 [18.6%]) (Table 1).

Age, sex, number of days from symptom onset to visit clinic and the number of people met did not differ between COVID-19 and patients without COVID-19. In patients with COVID-19, headache, dysgeusia, anosmia, cough and sputum production are significantly higher than in patients without COVID-19 (70.6% vs. 40.8%, 29.4% vs. 3.8%, 47.1% vs. 2.6%, 52.9% vs. 25.9% and 47.1% vs. 18.1%, respectively). Patients with COVID-19 visited nightclub, karaoke and izakaya or bar, and traveled on a train significantly more often than patients without COVID-19 (13.3% vs. 1.6%, 13.3% vs. 1.9%, 33.3% vs. 7.1%, 26.7% vs. 8.1%, respectively). In patients without COVID-19, there were no positive symptoms or place visited where people gathered in comparison with patients with COVID-19 (Table 1).

Comparison through separating each point at 37.0, 37.5, 38.0, 38.5, 39.0 degrees Celsius, respectively, showed that fever did not differ between COVID-19 and non-COVID-19 (Table 2).

### 3.2. Association between Possible Factors and COVID-19

Bivariate logistic regression analysis showed that nine variables (headache (OR, 3.48, 95% confidence interval (CI), 1.20–10.10), dysgeusia (OR, 10.58, CI, 3.25–34.47), anosmia (OR, 32.99, CI. 10.34–105.22), cough (OR, 3.21, CI, 1.20–8.58), sputum production (OR, 4.03, CI, 1.50–10.88), nightclub (OR, 9.01, CI, 1.61–50.31), karaoke (OR, 7.49, CI, 1.39–40.25), izakaya or bar (OR, 6.08 CI, 1.97–18.80) and train (OR, 3.91 CI, 1.19–12.89)) were significantly associated with COVID-19 (Table 3).

Multivariate logistic regression model using the nine variables as covariates showed that they were headache (OR, 3.54 CI, 0.93–13.48; *p* = 0.064), dysgeusia (OR, 0.44 CI, 0.05–4.19; *p* = 0.478), anosmia (OR, 29.3 CI, 4.17–205.61; *p* < 0.001), cough (OR, 1.90 CI, 0.47–7.67; *p* = 0.367), sputum production (OR, 2.67 CI, 0.70–10.19; *p* = 0.150), nightclub (OR, 0.86 CI, 0.02–41.13; *p* = 0.938), karaoke (OR, 0.62 CI, 0.01–32.21; *p* = 0.814), izakaya or bar (OR, 7.06 CI, 1.45–34.36; *p* = 0.016), train (OR, 0.39 CI, 0.029–5.24; *p* = 0.476). Then, stepwise backward regression (*p* < 0.1) improved the model; that is, using headache (OR, 3.31 CI, 0.98–11.20; *p* = 0.054), anosmia (OR, 25.94 CI, 7.15–94.14; *p* < 0.001), sputum production (OR, 3.32 CI, 1.01–10.90; *p* = 0.048) and izakaya or bar (OR, 4.23 CI, 0.99–18.03; *p* = 0.051), the AUC was 0.87 (CI, 0.769–0.971).

### 3.3. Development of the COVID-19 Scoring System

These factors were included in the COVID-19 positive score. Based on the multiple logistic analysis, we created the following formula to predict patients who have COVID-19.

*p* = 3 × headache + 26 × anosmia + 3 × sputum production + 4 × izakaya or bar, where *p* denotes the probability of positive COVID-19 real-time RT-PCR test. Using this formula, we calculated AUC, sensitivity, specificity and the overall rate of correct classification using the total score of COVID-19. The accuracy of COVID-19 positive was assessed using the AUC (Figure 1). The AUC of COVID-19 score was 0.870 (95% CI = 0.766 to 0.967), which indicates moderate accuracy of scale [13,14]. According to sensitivity, specificity and the overall rate of correct classification, the cut-off point of positive COVID-19 PCR was 7, in which sensitivity, specificity and the overall rate of correct classification were 76.5, 86.6, and 86.1%, respectively (Table 4). From these results, we developed the COVID-19 scoring system (Table 5).

## 4. Discussion

We developed a score to discriminate COVID-19 easily at the first visit to a general practitioner through only four factors. Multivariate analysis revealed that four factors, which were anosmia, headache, sputum production and izakaya or bar, discriminated between COVID-19 and non-COVID-19 cases. The score, which was a weighted sum by the strength of the association with COVID-19, showed moderate accuracy (AUC = 0.870, 95% CI = 0.766 to 0.967) [13,14].

Surprisingly, the presence of fever and dysgeusia was not useful to discriminate patients with COVID-19. Although patients with COVID-19 tend to have fever [15], it is not a useful indicator to discriminate COVID-19 from outpatients with common cold symptoms including fever. Although it has been reported that COVID-19 had various symptoms [16], this study showed that the identified four factors can be obtained by a simple interview or even questionnaire; it might make it possible to find patients with COVID-19 easily and quickly. Severity prediction scoring of COVID-19 has been reported; comorbidities, abnormal blood tests, age and dyspnea were the risk factors of severity [17,18]. However, there are not many reports of diagnostic prediction scoring. Some reports considered a scoring system using blood test data and chest CT to discriminate COVID-19 [6,19,20]. Similar to COVID-19, the influenza virus causes respiratory infections. The diagnostic scoring system of influenza reported chills, cough and muscle aches predicting influenza [21].

Comparing each case, about 100 people were examined, and the predictive power was all moderate (AUC = 0.81–0.86), similar to our study. Our study could not investigate blood tests and chest CT of all patients, but investigated in detail patient behavior, such as people who met and visited places. The different variables may cause differences in hospitals with rich facilities and clinics with poor facilities. Additionally, the risk factors of influenza differ from the risk factors of COVID-19 in this study and may help to discriminate COVID-19 from influenza based on clinical symptoms during both COVID-19 and influenza epidemics. On the other hand, interestingly, even in this influenza scoring system, symptoms such as fever and cough cannot be applied, suggesting that cold-like symptoms such as fever and cough may be less sensitive, including moderate patients. More detailed predictions may be expected by combining these reports with this study.

This study has several limitations. First, the real time RT-PCR test was performed only if the patient had one or more symptoms, close contact history or other factors which led to a suspicion of COVID-19 by the doctor. However, the test criteria may vary from doctor to doctor, and we did not include patients who did not undergo a real-time RT-PCR test. In addition, cases that were completely asymptomatic and were unaware of their contact history were excluded. Second, it is a single-center study of Japanese patients over a period of two weeks. Thus, the generalizability of the findings to other populations is limited. For example, COVID-19 has been reported to cause dysgeusia [5] or cough, but it was not significantly in this study. They are possibly a secondary symptom and may not be associated with COVID-19, but it is probable that the sample size was not adequate for a significant difference. Third, a patient who had visited a host club or hostess club, nightclub, pachinko parlor, karaoke, izakaya or bar might not inform us of their visit to them due to social stigma. For example, there was no significant difference in nightclub visits between COVID-19 cases and non-COVID-19 cases. However, since there are reported clusters in such places [10,22], the risk of infection may be considered. Selection bias may be present, as the association between risk factors and COVID-19 might be underestimated. Forth, severity of symptoms among patients with COVID-19 was not compared in this study. However, several symptoms such as fever and fatigues may be significant only in patients with severe COVID-19. Therefore, further studies to discriminate non-COVID-19, mild COVID-19 and severe COVID-19 are required in the future.

In conclusion, we have developed a score to easily discriminate COVID-19. This might be useful in clinical settings, especially in general practice. Further studies, in particular prospective studies confirming the usability of the score in different cultural settings and improving the accuracy of the instrument, are needed.

## Figures and Tables

**Figure 1 jcm-10-00854-f001:**
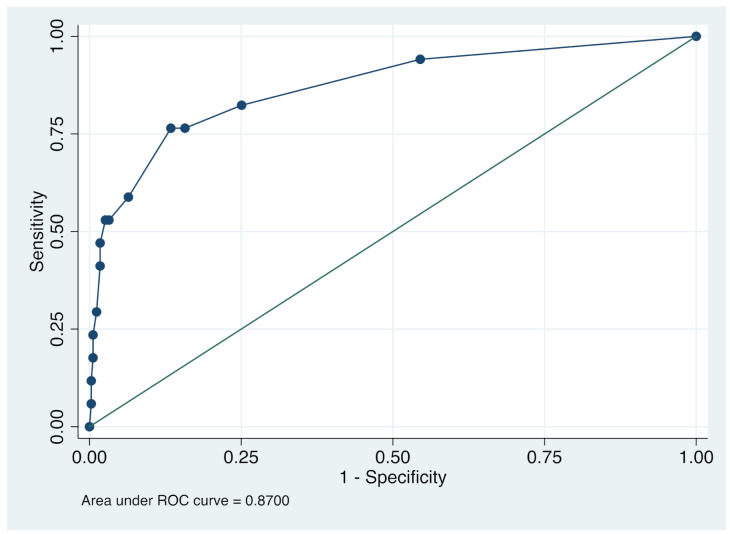
Receiver operating characteristic (ROC) curve in four independent cohorts.

**Table 1 jcm-10-00854-t001:** Demographic and clinical characteristics of patients who visited the clinic and underwent COVID-19 real-time RT-PCR test.

Characteristic	Total	COVID-19 Real-Time RT-PCR Test	
		Positive	Negative
Number	360	17 (4.7)	343 (95.3)
Age			
Mean (SD)	41.1 (18.8)	39.6 (14.3)	41.2 (19.0)
Range	1–94	20–71	1–94
≤19 (%)	35 (9.7)	0 (0)	35 (10.2)
20–29 (%)	77 (21.4)	5 (29.4)	72 (21.0)
30–39 (%)	54 (15.0)	4 (23.5)	50 (14.6)
40–49 (%)	89 (24.7)	2 (11.8)	85 (24.8)
50–59 (%)	41 (11.4)	2 (11.8)	39 (11.4)
60–69 (%)	33 (9.2)	0 (0)	33 (9.6)
70–79 (%)	21 (5.8)	2 (11.8)	19 (5.5)
80 < (%)	10 (2.8)	0 (0)	10 (2.9)
Sex			
Female (%)	176 (48.9)	7 (41.2)	169 (49.2)
Male (%)	184 (51.1)	10 (58.8)	174 (50.7)
Days from symptom onset to examination			
Mean (SD)	3.4 (6.1)	2.9 (1.8)	3.4 (6.2)
Range	0–74	0–6	0–74
People around patients			
Housemate (excluding the patient) (SD)	2.4 (2.1)	1.7 (1.3)	2.5 (2.2)
Work colleague in same space (SD)	15.5 (37.7)	7.8 (10.4)	15.9 (38.6)
People met within 1 week (SD)	20.9 (29.7)	17.6 (29.2)	21.0 (29.7)
People without mask met with 1 week (SD)	2.5 (6.4)	4.2 (5.7)	2.4 (6.4)
Symptom			
Body temperature (SD)	37.8 (1.0)	37.5 (0.8)	37.8 (1.0)
Headache (%)	152 (42.2)	12 (70.6)	140 (40.8)
Sore throat (%)	118 (32.8)	7 (41.2)	111 (32.4)
Dysgeusia (%)	18 (5.0)	5 (29.4)	13 (3.8)
Anosmia (%)	17 (4.7)	8 (47.1)	9 (2.6)
Nasal discharge (%)	61 (16.9)	5 (29.4)	56 (16.3)
Cough (%)	98 (27.2)	9 (52.9)	89 (25.9)
Sputum production (%)	70 (19.4)	8 (47.1)	62 (18.1)
Nausea/vomiting (%)	30 (8.3)	1 (5.9)	29 (8.5)
Diarrhea (%)	62 (17.2)	2 (11.8)	60 (17.5)
Stomachache (%)	38 (10.8)	1 (5.9)	37 (10.8)
Fatigue (%)	173 (48.1)	9 (52.9)	164 (47.8)
Shortness of breath (%)	42 (11.7)	2 (11.8)	40 (11.7)
Joint pain (%)	66 (18.3)	4 (23.5)	62 (18.1)
Myalgia (%)	58 (16.1)	4 (23.5)	54 (15.7)
Lack of appetite (%)	88 (24.4)	3 (17.6)	85 (24.8)
Visited place			
Close contact to patients with COVID-19 (%)	27 (7.5)	2 (13.3)	25 (8.1)
Host club and/or hostess club (%)	3 (0.8)	0 (0)	3 (1.0)
Nightclub (%)	7 (1.9)	2 (13.3)	5 (1.6)
Pachinko parlor (%)	9 (2.5)	1 (6.7)	8 (2.6)
Karaoke (%)	8 (2.2)	2 (13.3)	6 (1.9)
Izakaya or Bar (%)	27 (7.5)	5 (33.3)	22 (7.1)
Amusement arcade (%)	9 (2.5)	0 (0)	9 (2.9)
Movie theater (%)	7 (1.9)	0 (0)	7 (2.3)
Theater (%)	1 (0.3)	0 (0)	1 (0.3)
Live music club (%)	0 (0)	0 (0)	0 (0)
Amusement park (%)	10 (2.8)	0 (0)	10 (3.2)
Hospital (%)	84 (23.3)	3 (20.0)	80 (25.8)
Dental clinic (%)	13 (3.6)	0 (0)	13 (4.2)
Beauty salon (%)	27 (7.5)	1 (6.7)	26 (8.4)
Elderly day care center (%)	11 (3.1)	1 (6.7)	10 (3.2)
Sports gym (%)	9 (2.5)	0 (0)	9 (2.9)
Sports club (%)	10 (2.8)	1 (6.7)	9 (2.9)
Church (%)	0 (0)	0 (0)	0 (0)
Welfare facility (%)	19 (5.3)	0 (0)	19 (6.1)
Train (%)	29 (8.1)	4 (26.7)	25 (8.1)
Taxi (%)	9 (2.5)	1 (6.7)	8 (2.6)
Other place people gathered (%)	45 (12.5)	0 (0)	41 (13.2)
Outside the prefecture (%)	67 (18.6)	5 (33.3)	63 (20.3)
N/D (%)	6 (1.7)	0 (0)	6 (1.9)

SD, standard deviation.

**Table 2 jcm-10-00854-t002:** Number and percentage of patients having fever over each temperature (37.0, 37.5, 38.0, 38.5, 39.0), and crude odds ratios and 95% confidence intervals of risk factors for COVID-19 at each temperature, respectively.

Fever	Number		Total	Odds Ratio (95% CI)	*p* Value
	COVID-19	Non-COVID-19			
37 °C≤ (%)	12 (70.6)	271 (76.7)	283 (79.3)	0.61 (0.21–1.79)	0.37
37.5 °C≤ (%)	8 (47.1)	198 (58.2)	206 (57.7)	0.64 (0.24–1.69)	0.366
38 °C≤ (%)	6 (35.3)	142 (41.8)	148 (41.5)	0.76 (0.27–2.10)	0.598
38.5 °C≤ (%)	2 (11.8)	81 (23.8)	83 (23.2)	0.426 (0.10–1.90)	0.264
39 °C≤ (%)	1 (5.8)	49 (14.4)	50 (14.0)	0.371 (0.048–2.86)	0.342

CI, Confidence interval.

**Table 3 jcm-10-00854-t003:** Crude odds and adjusted odds ratios and 95% confidence intervals of risk factors for COVID-19.

Variables		Crude Model		Adjusted Model 1		Adjusted Model 2	
		Odds Ratio (95% CI)	*p* Value	Odds Ratio (95% CI)	*p* Value	Odds Ratio (95% CI)	*p* Value
Symptom	Headache	3.48 (1.20–10.10)	0.022	3.54 (0.93–13.48)	0.064	3.31 (0.98–11.20)	0.054
	Sore throat	1.46 (0.54–3.95)	0.452				
	Dysgeusia	10.58 (3.25–34.47)	<0.001	0.44 (0.05–4.19)	0.478		
	Anosmia	32.99 (10.34–105.22)	<0.001	29.3 (4.17–205.61)	<0.001	25.94 (7.15–94.14)	<0.001
	Nasal discharge	2.14 (0.72–6.30)	0.169				
	Cough	3.21 (1.20–8.58)	0.02	1.90 (0.47–7.67)	0.367		
	Sputum production	4.03 (1.50–10.86)	0.006	2.67 (0.70–10.19)	0.15	3.32 (1.01–10.90)	0.048
	Nausea/vomiting	0.68 (0.09–5.29)	0.710				
	Diarrhea	0.63 (0.14–2.82)	0.545				
	Stomachache	0.52 (0.07–4.01)	0.528				
	Fatigue	1.23 (0.46–3.26)	0.410				
	Shortness of breath	1.01 (0.22–4.58)	0.01				
	Joint pain	1.39 (0.44–4.42)	0.56				
	Myaglia	1.65 (0.52–5.24)	0.398				
	Lack of appetite	0.65 (0.18–2.32)	0.507				
Visited place	Close contact to patients with COVID-19	1.70 (0.37–7.84)	0.680				
	Host club and/or hostess club	N/A	N/A				
	Nightclub	9.01 (1.61–50.31)	0.012	0.86 (0.02–41.13)	0.938		
	Pachinko parlor	2.62 (0.31–22.21)	0.880				
	Karaoke	7.49 (1.39–40.25)	0.019	0.62 (0.01–32.21)	0.814		
	Izakaya or Bar	6.08 (1.97–18.80)	0.002	7.06 (1.45–34.36)	0.016	4.23 (0.99–18.03)	0.051
	Amusement arcade	N/A	N/A				
	Movie theater	N/A	N/A				
	Theater	N/A	N/A				
	Live music club	N/A	N/A				
	Amusement park	N/A	N/A				
	Hospital	1.01 (0.32–3.19)	0.984				
	Dental clinic	N/A	N/A				
	Beauty salon	0.76 (0.10–5.98)	0.796				
	Elderly day care center	2.08 (0.25–17.27)	0.497				
	Sports gym	N/A	N/A				
	Sports club	2.32 (0.28–19.43)	0.438				
	Church	N/A	N/A				
	Welfare facility	N/A	N/A				
	Train	3.91 (1.19–12.89)	0.025	0.39 (0.029–5.24)	0.476		
	Taxi	2.62 (0.31–22.210	0.378				
	Other place people gathered	2.27 (0.71–7.28)	0.169				
	Outside the prefecture	1.37 (0.43–4.33)	0.595				
	AUC (95% CI)			0.861 (0.75–0.97)		0.870 (0.77–0.97)	

CI, Confidence interval.

**Table 4 jcm-10-00854-t004:** Prediction parameters for COVID-19 total score.

Score	Sensitivity	Specificity	Overall Rate of Correct Classification	LR+	LR−
0	100.0	0.0	4.72	1.00	
3	94.1	45.5	47.8	1.73	0.13
4	76.5	84.3	83.9	4.86	0.28
6	76.5	86.6	86.1	5.70	0.27
7	58.8	93.6	91.9	9.17	0.44
10	52.9	97.4	95.3	20.18	0.48
26	47.1	97.4	95.0	17.93	0.54
29	41.2	98.3	95.6	23.54	0.60
30	23.5	99.4	95.8	40.35	0.77
32	17.6	99.4	95.6	30.26	0.83
33	11.8	99.7	95.6	40.35	0.88
36	5.9	99.7	95.3	20.18	0.94
37	0.0	100.0	95.3		1.00

**Table 5 jcm-10-00854-t005:** Development of COVID-19 scoring system.

Question	Answer	Score
1 Did you have a headache?	□ Yes	3
	□ No	0
2 Did you have anosmia?	□ Yes	26
	□ No	0
3 Did you have sputum?	□ Yes	3
	□ No	0
4 Did you visit a tavern within 1 week?	□ Yes	4
	□ No	0
Total score (≥7 indicates a high risk for COVID-19 positive)		( )

## Data Availability

The data presented in this study are available upon request from the corresponding author. The data are not publicly available due to privacy restrictions.

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
