# Peer review of "Validity of Clinical Symptoms Score to Discriminate Patients with COVID-19 from Common Cold Out-Patients in General Practitioner Clinics in Japan"

_jcm, 2021, doi:10.3390/jcm10040854_

Round 1

Reviewer 1 Report

Dear Authors,

my congratulations to this good presentation of your results!

To my opinion, this is an interesting summary of clinical findings and symptoms presented by patients positive or negative for Covid-19. Nevertheless, the availability and costs for PCR and antigen tests are no more an argument to decide whether to test a patient or not; please adapt the text in lines 55-57 accordingly. The findings you present are not new, but the presentation of a scoring system is a nice idea.

Please change:

Line 23: visited history - history of visiting

Line 28: Current study needs to be replicated - e.g.: The findings of this study need to be confirmed (or verified) ...

Line 109: ... on a previous study (10), which was carried out as follows: the score was the score was tripled when ...

I wish you all the best!

Reviewer 2 Report

Authors examined the records of 360 patients who visited a clinic with suspicion of infectious diseases and underwent COVID-19 PCR test between 1 and 14 August 2020. Authors investigated the clinical symptoms to discriminate between COVID-19 and non-COVID-19 cases.

Although this manuscript is potentially interesting, several issues arise.

Scoring system should be shown in Table.

Is dysgeusia or cough not important?

Is karaoke important?

Fever and fatigues are important for severe cases.

The limitation of this study should be stated.

The severity of patients with COVID-19 positive should be shown.

Reviewer 3 Report

A nice study that has clear aim and is of high relevance in an ongoing pandemic. The main limitation may in fact be the retrospective character and that not all patients were therefore tested, as mentioned by the authors. The score should be transferred to prospective study efforts.

Reviewer 4 Report

Dear Editor,

Sonoda and colleagues reviewed the main demographic characteristics, symptoms, signs, and social behavior of the patient of an outpatient practice in Japan that mainly serves patients with symptoms suggestive of a respiratory infection. The investigators compared these characteristics between patients with and without SARS-CoV-2 infection. Then, they attempted to create a model that could be used to differentiate between COVID-19 and non-SARS-CoV-2 viral syndromes. A model was created by using 4 variables that were found to be associated with COVID-19. This model was found to have moderate predictive power.

This goal was very ambitious for several reasons. We empirically know that symptoms of COVID-19 can be very similar to the ones of patients with non-SARS-CoV-2 infections. Therefore, many thousands of patients would be needed to detect small differences. One of the limitations of this study is the small sample size.

The investigators were not able to create a strong predictive model and, in my opinion, one of the messages of this study should be that rapid testing is the safest option and this is where we should focus.

Despite its limitation, this study has useful data and nice concept that deserve to become publicly available after several modifications are pursued.

Please see my comments below.

Introduction

*The introduction is nicely written

*Would site this study PMID: 32422233 and this study https://doi.org/10.1016/j.mayocp.2021.01.001 along with the current reference number 3. Both studies report, among others, the most common symptoms in COVID-19.

*Page 1, line 43: “COVID-19 infection”. There is COVID-19 (disease) and SARS-CoV-2 infection. Therefore, I would rephrase

*COVID-19 patients: would rephrase to patients with COVID-19

*The objective can be stated more clearly in the last paragraph of introduction

Methods

*Language edits are needed in this section

*Would suggest the authors to report which guidelines they followed to report the findings of this observational study (e.g., STROBE) and submit the relevant checklist as a supplementary file.

*Page 2, line 81: would change word “questions” to “variables”

Results

*Language edits are needed in this section, as well

*Do the authors mean “sputum production” by the word “sputum”?

Discussion:

*Would recommend the authors to better summarize the main findings of the study in the first paragraph of the discussion.

*Would suggestion to expand more on the discussion. For example, the second paragraph can be dedicated to the most common symptoms and signs and findings of other studies. It can also report the most common symptoms of other viral respiratory infection. The third paragraph can compare the scoring system the authors created with other scoring systems. The forth paragraph can be about strengths and limitations, and the last one can be the conclusion.  

Round 2

Reviewer 2 Report

Authors sufficiently responded my comments. I have no further comment. 

Reviewer 4 Report

The authors have substantially addressed my prior comments and concerns. 

In my opinion, the manuscript still needs a relatively thorough English language editing so that the authors' work is better presented. 

I highly recommend to assign a professional language editor